# LithoBench: Benchmarking AI Computational Lithography for Semiconductor Manufacturing

**Su Zheng**[1]    **Haoyu Yang**[2]    **Binwu Zhu**[1]    **Bei Yu**[1]    **Martin D.F. Wong**[3]
[1]The Chinese University of Hong Kong
[2]nVIDIA, Austin, USA
[3]Hong Kong Baptist University

## Abstract

Computational lithography provides algorithmic and mathematical support for resolution enhancement in optical lithography, which is critical for semiconductor manufacturing. The time-consuming lithography simulation and mask optimization processes limit the practical application of inverse lithography technology (ILT), a promising solution to the challenges of advanced-node lithography. Although machine learning for ILT has shown promise for reducing the computational burden, this field lacks a dataset that can train the models thoroughly and evaluate the performance comprehensively. To boost the development of AI-driven computational lithography, we present the LithoBench dataset, a collection of circuit layout tiles for deep-learning-based lithography simulation and mask optimization. LithoBench consists of more than 120k tiles that are cropped from real circuit designs or synthesized according to topologies of widely adopted ILT testcases. Ground truths are generated by a famous lithography model in academia and an advanced ILT method. We provide a framework to design and evaluate deep neural networks (DNNs) with the data. The framework is used to benchmark state-of-the-art models on lithography simulation and mask optimization. LithoBench is available at `https://github.com/shelljane/lithobench`.

## 1   Introduction

Semiconductor lithography transfers circuit patterns drawn on a mask onto a silicon wafer, which is essential for the fabrication of integrated circuits (IC). It typically accounts for about 30% of the cost of IC manufacturing. As transistor sizes continue to shrink, lithography tends to be the technical limiting factor for further advances [1]. Diffraction or process effects may distort the patterns on the wafer, leading to performance degradation and even failures. Given the importance, how to ensure the correctness of semiconductor lithography becomes a critical issue in industry and academia.

Optical proximity correction (OPC) is a technique used to improve the accuracy and quality of lithography patterns on semiconductor wafers. OPC evoles from rule-based methods [2] to model-based approaches [3–5], improving the resilience to manufacturing variation. Inverse lithography technology (ILT) [6–8] is a mathematically rigorous inverse approach that optimizes the mask to achieve the desired results on the wafer. It has been explored and developed as the next generation of OPC, promising a solution to challenges of advanced technology such as extreme ultraviolet (EUV).

However, there exist significant challenges that limit the broad application of ILT. A major reason is that ILT typically consumes a significant amount of runtime, making it challenging to implement ILT at a production-level scale. Moreover, the patterns generated by ILT algorithms may be too complex to be produced efficiently. To alleviate the problems above, DNN-based ILT algorithms have been proposed to reduce the runtime and complexity [9–17]. Compared with traditional methods, the improvement of DNN-based ILT algorithms has two aspects. Firstly, unlike traditional methods that

37th Conference on Neural Information Processing Systems (NeurIPS 2023) Track on Datasets and Benchmarks.

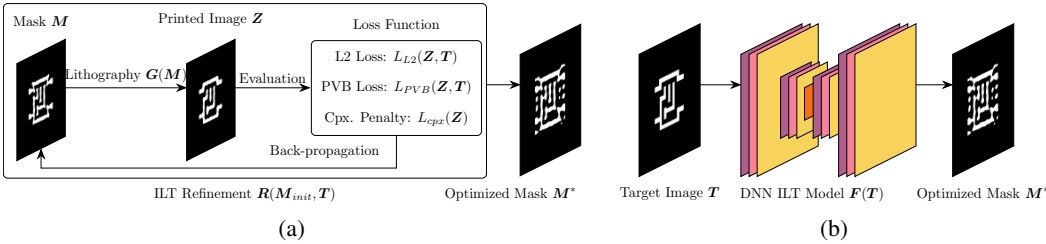

Figure 1: Overview of (a) traditional ILT and (b) DNN-based ILT.

require multiple iterations to optimize the mask, DNN-based methods can generate a mask in one forward pass. A few iterations of finetuning can further improve the quality of the mask. Secondly, by incorporating loss functions for complexity reduction, deep learning methods can remarkably improve the printability of ILT results [13, 14]. Deep learning shows great potential in advancing ILT.

Integrated circuits are represented by multi-layer layouts for manufacturing. A mask contains the patterns of one layout layer, optimized by ILT algorithms to produce desired printed patterns on silicon wafers. To achieve affordable runtimes, ILT algorithms typically generate the entire mask by optimizing fixed-size tiles cropped from the layout layer [18]. Fig. 1a presents a representative workflow of traditional ILT algorithms, which optimizes the mask iteratively. At each iteration, the lithography simulation model can output the printed image after the lithography processes. After that, the loss function evaluates the printed image in terms of error (L2 loss), process variation band (PVB), complexity (cpx.), etc. Given a loss function, the mask can be optimized by gradient descent [7] and other methods [19, 20]. Fig. 1b shows the DNN-based ILT flow. The input is the target image $\boldsymbol{T}$, which contains the target shapes that should be printed. The DNN ILT model outputs the optimized mask $\boldsymbol{M}^*$ in one forward pass, which is significantly faster than traditional ILT.

In recent years, researchers have proposed various DNN-based ILT methods [9–17], which can significantly boost the speed and performance of ILT with excellent initialization, refinement, and GPU acceleration. However, since the ILT methods are evaluated in different scenarios with different training data, it is difficult to compare them comprehensively and challenging to apply them in production. Specifically, circuit layouts typically consist of multiple layers of patterns. The shape, size, and density of the patterns vary from layer to layer. An ILT method may have only been evaluated on metal-layer [14] or via-layer [12] testcases. Nevertheless, a via layer typically contains square shapes, while a metal layer can include diverse rectilinear polygons. It is questionable to assert that one of the methods above is better because they are not evaluated in the same scenarios. In addition, some models are trained on synthetic data [9], while others focus on real-world designs [21]. Therefore, a representative and comprehensive dataset should cover metal and via layers, as well as synthetic and real-world designs. It is valuable to construct a common dataset that can train and evaluate DNN ILT models in various scenarios.

To evaluate ILT methods, multiple metrics should be considered. For example, L2 loss measures the difference between the printed patterns and the target shapes. PVB estimates the robustness of the mask against process variations. Shot count shows the complexity of the mask, which affects the manufacturing time and cost. Runtime is also an important factor. To choose an appropriate algorithm, users usually need to make a trade-off between the metrics. As a result, there is a need for a platform that can effectively benchmark various ILT methods using comprehensive metrics.

DNN can be employed in ILT for lithography simulation or mask optimization. In simple terms, the output of ILT is an optimized mask $\boldsymbol{M}^*$, which can get similar results as the target patterns $\boldsymbol{T}$ after the lithography processes $\boldsymbol{G}(\boldsymbol{M}^*)$. Existing works [22, 23] approximate the lithography processes $\boldsymbol{G}(\boldsymbol{M}^*)$ with DNN-based lithography simulation models, which can not only provide faster evaluation of the mask quality but also be used in mask refinement. DNN-based mask optimization models [9, 11, 12, 16] are designed to directly obtain the optimized mask, formulated by $\boldsymbol{M}^* = \boldsymbol{F}(\boldsymbol{T})$.

To meet the above requirements, we present LithoBench, a collection of circuit layout tiles to train DNN models for lithography simulation and mask optimization. Its benchmarking platform can provide an extensive evaluation of the models. Our contributions are summarized as follows.

- We assemble a dataset for DNN-based lithography simulation and mask optimization, which not only contains synthetic and real-world layout tiles but also covers metal and via layers. It is the first large-scale dataset for computational lithography built from real-world designs.
- We implement a platform to evaluate the DNN models for lithography simulation and mask optimization. Users can design and select appropriate models with our platform by benchmarking them in terms of quality, robustness, complexity, runtime, etc.
- We train and evaluate state-of-the-art (SOTA) DNN models for lithography simulation and mask optimization on LithoBench, providing a comprehensive assessment of them.

## 2 Related Work

This section provides the fundamentals of ILT. We begin by discussing the commonly used lithography simulation method. Next, we describe typical ILT methods that solve mask optimization via numerical optimization. Then we review the latest advancements in DNN models for lithography simulation and mask optimization. Finally, LithoBench is compared to existing datasets about lithography.

### 2.1 Lithography Simulation

The lithography simulation targets to approximate the real lithography process in chip manufacturing and provides an accurate estimation of the manufactured designs on silicon wafers. Lithography simulation consists of the optical projection and photoresist models. In optical projection, incident light passes through the mask, transmitting the spatial information of the mask patterns $M$ to the optical projection system. This results in the transformation of the input light intensity distribution into an aerial intensity distribution on the wafer plane. The intensity distribution is commonly represented by an aerial image $I$. The process can be modeled by Hopkins' diffraction theory [24]:

$$I = H\left(M\right) = \sum_{k=1}^{K} \mu_k |h_k \otimes M|^2,\tag{1}$$

where $h_k$ is the $k$th optical kernel function and $\mu_k$ is the corresponding weight. The notation $\otimes$ stands for convolution operation. $|\cdot|^2$ gets the squared modulus of each element.

After optical projection, a photoresist model transfers the aerial image $I$ to the printed image $Z$, which is also called resist image. To enable gradient descent in ILT algorithms, researchers commonly design the photoresist model as:

$$Z(x,y) = \sigma_Z\left(I(x,y)\right) = \frac{1}{1 + e^{(-\alpha(I(x,y)-I_{th}))}},\tag{2}$$

where $I_{th}$ is the intensity threshold, $\alpha$ is a constant number that controls the steepness of the function, and $(x,y)$ represents a coordinate on the aerial or resist image. Lithography simulation transforms the mask $M$ to the resist image $Z$ with the optical projection and photoresist models.

### 2.2 Mask Optimization via ILT

The application of ILT [25–32] in industrial productions began in 2005 [33–38]. It tries to solve the mask pattern such that after the lithography process, the remaining pattern on the silicon wafer is as close as the original design. Thus, ILT can be viewed as the inverse process of lithography simulation. In recent years, the ICCAD-13 benchmark [39] has facilitated extensive research on ILT in academia, such as MOSAIC [7], MultiLevel [40], GPU-LevelSet [8, 19], etc.

An ILT algorithm usually optimizes a parameter matrix $P$ that can be transformed to the mask image $M$ with the following function:

$$M(x,y) = \sigma_M(P(x,y)) = \frac{1}{1 + e^{(-\beta(P(x,y)-\gamma))}},\tag{3}$$

where $\beta$ is the constant steepness factor and $\gamma$ is a constant offset. With this function, the values in $M$ can be limited in $[0,1]$, while the values in $P$ have no limitation. Finally, the transformation from the parameter matrix $P$ to the resist image $Z$ can be formulated as:

$$Z = \sigma_Z\left(H\left(\sigma_M\left(P\right)\right)\right).\tag{4}$$

In mask manufacturing, the printed patterns on the wafer may differ drastically under varying process conditions (e.g. depth of focus, intensity of incident light). ILT algorithms typically consider three process corners, maximum, nominal, and minimum. Specifically, three series of optical kernels are employed to form the optical projection models $\boldsymbol{H}_{max}$, $\boldsymbol{H}_{nom}$, and $\boldsymbol{H}_{min}$. The corresponding resist images can be denoted by $\boldsymbol{Z}_{max}$, $\boldsymbol{Z}_{nom}$, and $\boldsymbol{Z}_{min}$.

L2 loss measures the difference between the nominal resist image and the target image, defined as:

$$L2(\boldsymbol{Z}_{nom}, \boldsymbol{T}) = \|\boldsymbol{Z}_{nom} - \boldsymbol{T}\|_2^2. \tag{5}$$

To enhance the robustness against varying process conditions, ILT algorithms are commonly required to reduce the process variation band (PVB) defined as:

$$PVB(\boldsymbol{Z}_{max}, \boldsymbol{Z}_{min}) = \|\boldsymbol{Z}_{max} - \boldsymbol{Z}_{min}\|_2^2. \tag{6}$$

ILT loss functions are usually based on $L2(\boldsymbol{Z}_{nom}, \boldsymbol{T})$, $PVB(\boldsymbol{Z}_{max}, \boldsymbol{Z}_{min})$, and additional components like complexity losses. Given the loss function, the parameter matrix $\boldsymbol{P}$ can be optimized by gradient descent. The final parameter matrix $\boldsymbol{P}^*$ can be binarized to obtain the optimized mask $\boldsymbol{M}^*$ by checking whether each element satisfies $\boldsymbol{\sigma}_M(\boldsymbol{P}^*(x, y)) > 0.5$.

### 2.3 DNN-based Methods

For lithography simulation, conditional generative adversarial network (CGAN) [41, 42] is utilized in LithoGAN [22], DAMO [12], and TEMPO [43]. Fourier Neural Operator (FNO) [44] inspires DOINN [23]. These works suggest that lithography simulation via DNN models is a rapidly developing field. To distinguish the lithography model based on Hopkins' diffraction theory from the DNN-based models, we refer to the theory-based one as the reference lithography simulation model.

In mask optimization, DNN-based methods also achieve tremendous success. For instance, GAN-OPC [9] trains a CGAN guided by reference optimized masks and lithography simulation for mask optimization. Its training scheme inspires subsequent research, such as Neural-ILT [11] and DAMO [12]. CFNO [16] presents a powerful model inspired by Vision Transformer [45] and FNO.

### 2.4 Lithography Datasets

ICCAD-13 [39] is a famous benchmark for mask optimization consisting of 10 $2\mu m \times 2\mu m$ metal-layer clips. However, it is targeting numerical optimization solutions only and the 10 instances are too small to fit AI solutions. GAN-OPC [9] releases around 4k synthetic tiles for metal-layer mask optimization. However, its size is small for a thorough training and the quality of the optimized masks falls behind SOTA results. Furthermore, it does not provide real-world designs and via-layer tiles, which limits its applications. Another limitation of existing datasets is that they do not support DNN-based lithography simulation, which is also a critical step in computational lithography. To overcome these weaknesses, we propose LithoBench, the first comprehensive dataset that simultaneously supports lithography simulation and mask optimization. Unlike previous datasets, LithoBench includes abundant and diverse data that covers synthetic and real-world layout tiles, as well as metal and via layers. The ground truths are generated by SOTA method, providing the high-quality data.

ICCAD-12 [46] is a dataset for lithography hotspot detection [47] (HSD), which aims to find the locations on the mask that may lead to defects on the printed patterns. HSD only indicates the presence of defects in certain regions, but provides little information for mask optimization. LithoBench focuses on lithography simulation, which is more important than HSD, providing detailed information about the printed patterns. Besides, AI approaches for HSD have been deeply studied in literature and are mature in production flows. Therefore, HSD-related benchmarks are not the scope of this paper.

## 3 Dataset

LithoBench consists of 133,496 tiles that forms four subsets: MetalSet, ViaSet, StdMetal, and StdContact. MetalSet and ViaSet are large subsets primarily used for training, compatible with existing research. StdMetal and StdContact are small subsets for evaluating the generalization ability of the models. In each subset, we prepare the target images, optimized masks, aerial images, and printed images. In lithography simulation tasks, the input of the DNN model is an optimized mask, and the outputs include an aerial image and a printed image. In mask optimization tasks, the input and output are a target image and the optimized mask, respectively.

Table 1: Summary of LithoBench

| Task | Lithography Simulation | | | | Mask Optimization | | | |
|------|---------|--------|----------|------------|---------|--------|----------|------------|
| Subsets | MetalSet | ViaSet | StdMetal | StdContact | MetalSet | ViaSet | StdMetal | StdContact |
| Training Tiles | 14,824 | 104,733 | 0 | 163 | 14,824 | 104,733 | 0 | 163 |
| Testing Tiles | 1,648 | 11,642 | 271 | 165 | 10 | 10 | 271 | 165 |

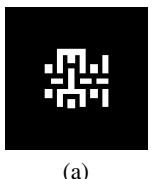 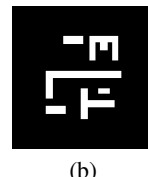 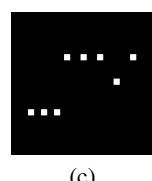 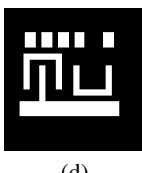 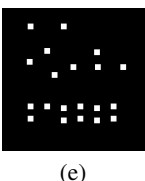

| (a) | (b) | (c) | (d) | (e) |

Figure 2: Samples from (a) ICCAD-13 [39], (b) MetalSet, (c) ViaSet, (d) StdMetal, (e) StdContact.

## 3.1 Data Collection

### 3.1.1 MetalSet

ICCAD-13 benchmark [39] is a famous benchmark for metal-layer mask optimization, used in various SOTA ILT methods [7–9, 11, 13–15, 19]. It contains 10 tiles in GLP format [39] that come from $32nm$ industrial layouts. We design MetalSet to enable the training of DNN-based model that can achieve high performance on ICCAD-13 benchmark. We synthesize 16,472 tiles using the layout generation method in [48]. Each tile is randomly generated following the design rules of ICCAD-13 benchmark. The rules refer to the limitations on shape widths, distances, and areas, determined by the technology node. The original ICCAD-13 benchmark is used to evaluate DNN-based mask optimization models that are trained on MetalSet. This setting is compatible with existing methods [9, 11, 13, 14]. Fig. 2a and Fig. 2b show examples of ICCAD-13 benchmark and MetalSet.

### 3.1.2 ViaSet

For via-layer ILT, we generate the layouts at the $45nm$ technology node with the IC design tool, OpenROAD [49, 50]. We select the `gcd` and `aes` circuits from the OpenROAD project and get their layouts in GDSII format, the de facto industry standard for electronic design automation. The Python package `python-gdsii` [51] is used to extract the shapes on the first via layers of the circuits and save them in the GLP format. We crop the layouts into $2048 \times 2048nm^2$ tiles with a stride of $512nm$. Following existing ILT methods, we only include the shapes within the central region, which has a size of $1280 \times 1280nm^2$. Finally, we get 116,415 clips of the layout. Following the MetalSet, we select 10 tiles with different complexity for testing.

### 3.1.3 StdMetal and StdContact

StdMetal and StdContact subsets include the layout tiles of the Nangate $45nm$ standard cells [52]. Nangate $45nm$ is an open IC library that is widely used in academia. A standard cell is a group of transistors and interconnect structures that provide a boolean logic function (e.g., AND, OR, XOR, XNOR, inverters) or a storage function (flip-flop or latch). Since standard cells are typically the basic building blocks of digital IC, the ILT of standard cells is important for manufacturing. We crop the layouts of the standard cells in the same way as ViaSet.

The StdMetal subset consists of the tiles on the first metal layer. Since the standard cells do not use via layers, we use the contact layer to build the StdContact subset, which also contains square shapes like the via layer. The 271 tiles in StdMetal are used to evaluate the generalization ability of models trained on MetalSet. Fig. 2d shows an example from StdMetal. On StdContact, we test the models trained on ViaSet with 165 tiles. We allow the finetuning on another 163 tiles because the density of shapes in StdContact is obviously higher than the density in ViaSet. Fig. 2c and Fig. 2e show the examples from ViaSet and StdContact, which can illustrate the difference in density.

## 3.2 Data Preparation

**Target Images**  We parse the data in the GLP format and put the shapes in the central region of a $2048 \times 2048$ image. Each pixel in the image corresponds to a $1 \times 1nm^2$ area of the circuit layout.

**Optimized Masks**  We employ an ILT method inspired by [14, 40] to get the ground truths. The lithography simulation model is given by the ICCAD-13 benchmark, which is for the $32nm$ technology node. Since the resolution required by $45nm$ is lower than $32nm$, it is also applicable for $45nm$ designs. In the ILT forward pass, we apply average pooling, (3), (1), and (2) to the parameters $\boldsymbol{P}$:

$$\boldsymbol{Z} = \boldsymbol{\sigma}_Z \left( \boldsymbol{H} \left( \boldsymbol{\sigma}_M \left( AvgPool \left( \boldsymbol{P} \right) \right) \right) \right). \tag{7}$$

Considering three process corners $\boldsymbol{H}_{max}$, $\boldsymbol{H}_{nom}$, and $\boldsymbol{H}_{min}$, we can obtain three resist images $\boldsymbol{Z}_{max}$, $\boldsymbol{Z}_{nom}$, and $\boldsymbol{Z}_{min}$. The loss function is defined as:

$$L_f(\boldsymbol{Z}_{nom}, \boldsymbol{Z}_{max}, \boldsymbol{Z}_{max}, \boldsymbol{T}) = \|\boldsymbol{Z}_{max} - \boldsymbol{T}\|_2^2 + \|\boldsymbol{Z}_{max} - \boldsymbol{Z}_{min}\|_2^2 + L_{curv}(\boldsymbol{Z}_{nom}), \tag{8}$$

where $\|\boldsymbol{Z}_{max} - \boldsymbol{T}\|_2^2$ and $\|\boldsymbol{Z}_{max} - \boldsymbol{Z}_{min}\|_2^2$ are for the minimization of L2 and PVB, respectively. $L_{curv}(\boldsymbol{Z}_{nom})$ is the curvature loss that can improve the smoothness of the mask [14]. In the backpropagation, we update the parameters $\boldsymbol{P}$ via gradient descent. Inspired by multi-level ILT [40], we optimize the mask at resolutions $256 \times 256$, $512 \times 512$, and $1024 \times 1024$ for 200, 100, and 100 iterations, respectively. Finally, we get a $2048 \times 2048$ mask by interpolating the result.

**Aerial and Printed Images**  An aerial image $\boldsymbol{I}$ is generated by applying (1) to the optimized mask $\boldsymbol{M}^*$ with the nominal kernel. A printed image $\boldsymbol{Z}$ is obtained by binarizing $\boldsymbol{I}$ with $I_{th}$.

## 3.3 Tasks

### 3.3.1 Lithography Simulation

Given an optimized mask, the lithography simulation model outputs the corresponding aerial and printed image. For MetalSet and ViaSet, we use $90\%$ of the data as the training set and the rest as the test set. The models trained on MetalSet can be tested on StdMetal without finetuning. The models trained on ViaSet are tested on StdContact after the finetuning on a few data.

Mean squared error (MSE) is adopted to measure the performance of lithography simulation. Furthermore, intersection over union (IOU) and pixel accuracy (PA) [23] are utilized to evaluate the quality of the predicted printed image. The IOU metric can be defined as:

$$\text{IOU}(\boldsymbol{Z}, \boldsymbol{T}) = \frac{\boldsymbol{Z}_1 \cap \boldsymbol{T}_1}{\boldsymbol{Z}_1 \cup \boldsymbol{T}_1}, \tag{9}$$

where $\boldsymbol{Z}_1$ and $\boldsymbol{T}_1$ are the regions in printed patterns and target shapes, respectively. PA is defined as:

$$\text{PA}(\boldsymbol{Z}, \boldsymbol{T}) = \frac{\boldsymbol{Z}_1 \cap \boldsymbol{T}_1}{\boldsymbol{T}_1}. \tag{10}$$

Note that we interpolate the output images to $2048 \times 2048$ before computing any metric.

### 3.3.2 Mask Optimization

For mask optimization, the model predicts a well-optimized mask according to the target image. The training splits used for mask optimization are the same as for the lithography simulation task. However, the test data of MetalSet and ViaSet are different. We use the 10 testcases in the ICCAD-13 benchmark to test the model on MetalSet. For ViaSet, we select 10 representative tiles as the testcases. StdMetal and StdContact provide hundreds of challenging samples for testing.

Mask optimization models aim to produce high-quality ILT results, not just mimic the reference optimized masks. We evaluate these metrics [7, 11] after binarizing the masks and printed images:

1. *L2* loss, as defined in (5), is a metric that measures the difference between the nominal resist image and the target image. Fig. 3a visualizes the *L2* loss.

2. *PVB*, as defined in (6), evaluates the robustness of the mask against varying process conditions. Fig. 3b visualizes the *PVB* between the maximum and minimum process corners.

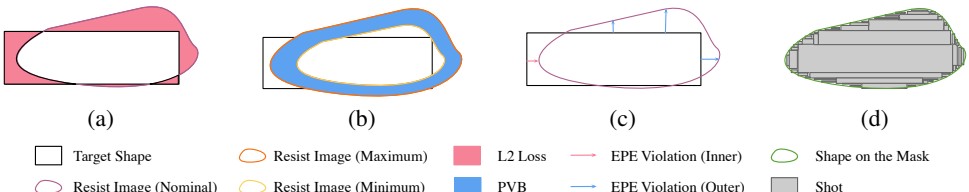

Figure 3: Illustration of the metrics. (a) *L2* measures the difference between the printed and target images. (b) *PVB* quantifies the maximum discrepancy between process corners. (c) *EPE* estimates the distortion of the printed image. (d) *#Shots* counts the rectangles needed to construct the mask.

Table 2: Comparison on Lithography Simulation

| Subtask | LithoGAN [22] $MSE_A$ | $MSE_P$ | IOU | PA | DAMO [12] $MSE_A$ | $MSE_P$ | IOU | PA | DOINN [23] $MSE_A$ | $MSE_P$ | IOU | PA | CFNO [16] $MSE_A$ | $MSE_P$ | IOU | PA |
|---|---|---|---|---|---|---|---|---|---|---|---|---|---|---|---|---|
| 1 | $9.8 \cdot 10^{-4}$ | $1.7 \cdot 10^{-2}$ | 0.38 | 0.43 | $\mathbf{8.4 \cdot 10^{-6}}$ | $7.5 \cdot 10^{-4}$ | **0.97** | **0.98** | $8.5 \cdot 10^{-6}$ | $\mathbf{6.6 \cdot 10^{-4}}$ | **0.97** | **0.98** | $1.9 \cdot 10^{-5}$ | $1.5 \cdot 10^{-3}$ | 0.94 | 0.97 |
| 2 | $2.6 \cdot 10^{-4}$ | $1.4 \cdot 10^{-3}$ | 0.47 | 0.53 | $3.0 \cdot 10^{-6}$ | $1.5 \cdot 10^{-4}$ | 0.94 | 0.96 | $\mathbf{1.9 \cdot 10^{-6}}$ | $\mathbf{1.0 \cdot 10^{-4}}$ | **0.96** | **0.98** | $3.8 \cdot 10^{-6}$ | $2.1 \cdot 10^{-4}$ | 0.92 | 0.96 |
| 3 | $1.4 \cdot 10^{-3}$ | $2.6 \cdot 10^{-2}$ | 0.30 | 0.34 | $2.5 \cdot 10^{-5}$ | $1.5 \cdot 10^{-3}$ | 0.95 | 0.97 | $\mathbf{1.8 \cdot 10^{-5}}$ | $\mathbf{1.2 \cdot 10^{-3}}$ | **0.96** | **0.98** | $2.6 \cdot 10^{-5}$ | $2.3 \cdot 10^{-3}$ | 0.93 | 0.96 |
| 4 | $2.7 \cdot 10^{-3}$ | $1.2 \cdot 10^{-2}$ | 0.01 | 0.01 | $4.6 \cdot 10^{-5}$ | $1.6 \cdot 10^{-3}$ | 0.87 | 0.93 | $2.4 \cdot 10^{-5}$ | $\mathbf{1.3 \cdot 10^{-3}}$ | **0.90** | **0.94** | $2.1 \cdot 10^{-5}$ | $2.2 \cdot 10^{-3}$ | 0.83 | 0.90 |
| Average | $1.3 \cdot 10^{-3}$ | $1.4 \cdot 10^{-2}$ | 0.29 | 0.33 | $2.1 \cdot 10^{-5}$ | $1.0 \cdot 10^{-3}$ | 0.93 | 0.96 | $\mathbf{1.3 \cdot 10^{-5}}$ | $\mathbf{8.2 \cdot 10^{-4}}$ | **0.95** | **0.97** | $1.7 \cdot 10^{-5}$ | $1.5 \cdot 10^{-3}$ | 0.91 | 0.95 |
| Runtime | **0.013** s / image | | | | 0.030 s / image | | | | 0.017 s / image | | | | 0.035 s / image | | | |

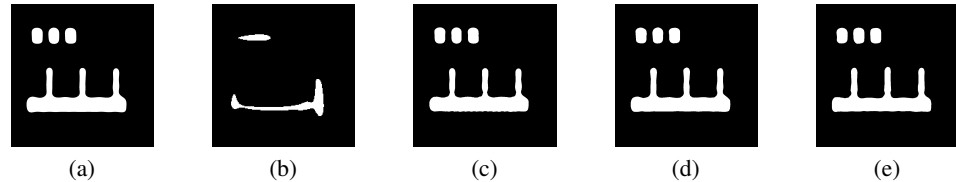

Figure 4: Lithography simulation. (a)Ground truth,(b)LithoGAN,(c)DAMO,(d)DOINN,(e)CFNO.

3. As shown in Fig. 3c, edge placement error (EPE) estimates the distortion of the resist image. We sample probe points equidistantly on horizontal and vertical edges of the target patterns. If the distance from the target pattern to the printed pattern is larger than the EPE constraint, it produces an EPE violation. The number of EPE violations is the *EPE* score of the mask.

4. Shot count (*#Shots*) is the number of rectangular shots for replicating the shapes on the mask. It can evaluate the complexity of an optimized mask. Fig. 3d shows an example, where a shape is fitted by rectangular shots. In LithoBench, we compute the shot count using adaptive rectangular decomposition [53] with `adaptive-boxes` [54] (MIT license).

Table 1 summarizes the statistics of LithoBench.

## 4 Experiments

### 4.1 Benchmarked Models

#### 4.1.1 Lithography Simulation Models

Multiple SOTA models are implemented using PyTorch [55] and OpenILT [56] on RTX3090 GPU. For lithography simulation, we implement LithoGAN [22], DAMO [12], DOINN [23], and CFNO [16]. The last layer of each model outputs two channels as the aerial and printed images. These models minimize the MSE between their outputs and the ground truths.

**LithoGAN** A CGAN is employed to fit the reference lithography simulation model. The generator of the CGAN is a fully convolutional network (FCN) [57] and the discriminator is a convolutional neural network (CNN). Images are downsampled to $256 \times 256$ and input to the model. Note that different models may vary in image resolution to make trade-offs between accuracy and runtime.

Table 3: Comparison on Mask Optimization

| Subtask | GAN-OPC [9] | | | | Neural-ILT [11] | | | | DAMO [12] | | | | CFNO [16] | | | |
|---|---|---|---|---|---|---|---|---|---|---|---|---|---|---|---|---|
| | $L_2$ | PVB | EPE | Shots | $L_2$ | PVB | EPE | Shots | $L_2$ | PVB | EPE | Shots | $L_2$ | PVB | EPE | Shots |
| 1 | 43414 | 41290 | 8.7 | 574 | 36670 | 42666 | 7.3 | 476 | **32579** | 41173 | **5.4** | 523 | 47814 | 46131 | 12.5 | **302** |
| 2 | 14767 | **6686** | 8.3 | **166** | 12723 | 8537 | 6.2 | 263 | **5081** | 9962 | **0.0** | 176 | 8949 | 9890 | 0.1 | 184 |
| 3 | 25929 | 23715 | 4.6 | 457 | 20045 | **23548** | 2.4 | 373 | **16120** | 23796 | **0.2** | 418 | 26809 | 26814 | 4.2 | **232** |
| 4 | 81378 | **4931** | 73.2 | 276 | **25422** | 41537 | **3.2** | 265 | 50445 | 35673 | 26.7 | 458 | 70740 | 17950 | 55.1 | 396 |
| Average | 41372 | **19156** | 23.7 | 368 | **23715** | 29072 | **4.8** | 344 | 26056 | 27651 | 8.0 | 394 | 38578 | 25196 | 18.0 | **279** |
| Runtime | **0.010** s / image | | | | 0.025 s / image | | | | 0.028 s / image | | | | 0.040 s / image | | | |

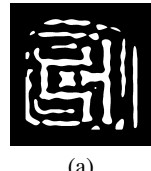 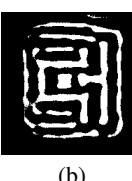 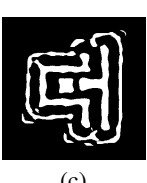 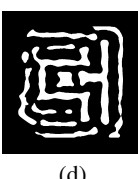 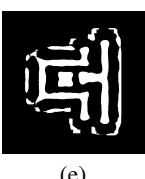

(a)      (b)      (c)      (d)      (e)

Figure 5: Mask optimization. (a)Ground truth,(b)GAN-OPC,(c)Neural-ILT,(d)DAMO,(e)CFNO.

**DAMO** It improves the CGAN for lithography simulation with the backbone based on UNet++ [58] and the multiscale discriminator inspired by pix2pixHD [59]. The resolution of DAMO is $1024 \times 1024$, which is significantly higher than LithoGAN.

**DOINN** Inspired by Fourier Neural Operator (FNO) [60], DOINN utilizes a novel reduced FNO architecture to fit the lithography simulation model. Its resolution is $1024 \times 1024$.

**CFNO** Combining the principles of Vision Transformer (ViT) [45] and FNO, the CFNO module is designed for efficient global layout embedding and resolving stitching issues caused by long-range dependency. The resolution for CFNO is also $1024 \times 1024$.

### 4.1.2 Mask Optimization Models

For mask optimization, we implement GAN-OPC [9], Neural-ILT [11], DAMO [12], and CFNO [16].

**GAN-OPC** CGAN also inspires the design of GAN-OPC. The training of GAN-OPC consists of two stages. In the first stage called ILT-guided pretraining, we train the generator to minimize the MSE between its outputs and the optimized masks. The second stage is based on the training process of GAN. During the training of the generator, we use the L2 loss formulated by (5) as an additional objective, where the printed image $\boldsymbol{Z}_{nom}$ is obtained by the reference lithography simulation model. GAN-OPC uses a resolution of $256 \times 256$.

**Neural-ILT** A UNet [61] is utilized in Neural-ILT to predict the optimized mask. Neural-ILT also has a pretraining stage like GAN-OPC. In the second stage, it uses $L2(\boldsymbol{Z}_{nom}, \boldsymbol{T}) + PVB(\boldsymbol{Z}_{max}, \boldsymbol{Z}_{min})$ as the objective function. The resolution of Neural-ILT is $512 \times 512$.

**DAMO** and **CFNO** use the same architectures as in lithography simulation.

### 4.1.3 Subtasks

For lithography simulation and mask optimization, we benchmark the models with the following subtask: (1) training on MetalSet, testing on MetalSet; (2) training on ViaSet, testing on ViaSet; (3) training on MetalSet, testing on StdMetal; (4) training on ViaSet, testing on StdContact. The last two are more challenging since high generalization ability is required in these subtasks.

## 4.2 Results on Lithograph Simulation

Table 2 presents the performance of the tested models on all subtasks. $MSE_A$ and $MSE_P$ are the MSE losses of the aerial and printed images. Fig. 4 shows the outputs images of the models. DOINN achieves the best performance with a competitive runtime. DAMO and CFNO have comparable performance, which is beyond the reach of LithoGAN. As shown in Fig. 4b, even a coarse generation of the target shapes is too difficult for LithoGAN. This can be explained by two factors. First, LithoGAN

uses a lower resolution than other methods, which makes it harder for LithoGAN to approximate the shapes in a higher resolution. Second, the convolution layers in LithoGAN progressively shrink the feature maps until they reach a size of $512 \times 1 \times 1$ at the intermediate layer. Since the printed image resembles the input mask, it is difficult to decode the complex shapes from such low-resolution feature maps. Thus, high-resolution features and outputs are essential for accurate lithography simulation.

## 4.3   Results on Mask Optimization

Table 3 compares the mask optimization models on all subtasks. Fig. 5 shows the output masks of the models. While GAN-OPC has the best PVB among these methods, it sacrifices L2 and EPE. It fails to learn the complicated curves that improve the L2 loss due to its low resolution. With a moderate resolution, Neural-ILT achieves moderate performance on the first three tasks. But it generalizes well to StdContact, enabling it to achieve the best average L2 and EPE. DAMO has the best performance on the first three tasks, while CFNO achieves lower mask complexity due to the small shot counts. In mask optimization, the model designer may consider the trade-off between resolution, generalization, and complexity. It is challenging to design a model that is optimal across all metrics.

## 5   Conclusion

In this paper, we present LithoBench, a dataset for DNN-based lithography simulation and mask optimization with an evaluation platform. LithoBench includes 133,496 layout tiles that involve the metal and via layers of circuit layouts. It can not only support the typical setup where a model is trained and evaluated on similar tiles, but also evaluate the model's generalization ability to unseen data. Given the data, state-of-the-art DNN models for lithography simulation and mask optimization are trained and evaluated on LithoBench to show their advantages and weaknesses. We hope LithoBench can contribute to the further development of computational lithography.

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
