# LithoBench: Benchmarking AI Computational Lithography for Semiconductor Manufacturing Supplementary Materials

**Su Zheng**[1]    **Haoyu Yang**[2]    **Binwu Zhu**[1]    **Bei Yu**[1]    **Martin D.F. Wong**[3]
[1]The Chinese University of Hong Kong
[2]nVIDIA, Austin, USA
[3]Hong Kong Baptist University

## A    Functionalities Provided by LithoBench

In addition to the data and data loaders, LithoBench also provides functionalities that can facilitate the development of DNN-based and traditional ILT algorithms. Based on PyTorch [1] and OpenILT [2], we implement the reference lithography simulation model as a PyTorch module, which can be used like a DNN layer. The GPU-based fast Fourier transform (FFT) can boost the speed of lithography simulation. PyTorch optimizers can be directly employed to optimize the masks according to ILT loss functions, significantly simplifying the development of ILT algorithms.

To evaluate ILT results, LithoBench provides a simple interface to measure the L2 loss, PVB, EPE, and shots of the output masks. It also incorporates Python programs that can train and test the models mentioned in this paper. We provide the base classes of lithography simulation and mask optimization models. By inheriting the classes, users can easily build their own models that can be trained and tested by LithoBench, without the need of writing the code for data loading and evaluation. Fig. 1 shows a typical flow for training and evaluating an ILT model. The users only need to implement the model and the five functions, i.e. pretrain, train, save, load, run. We include a pretraining interface to support the commonly adopted two-stage training scheme. However, pretraining is optional since methods like DOINN do not use two-stage training. Similar interfaces are required for lithography simulation.

## B    Reference ILT Algorithm

The reference ILT algorithm generates the optimized masks in LithoBench, which can be utilized to guide the pretraining or training of DNN-based mask optimization models. The forward pass of our reference ILT algorithm is:

$$\boldsymbol{Z} = \boldsymbol{\sigma}_Z \left( \boldsymbol{H} \left( \boldsymbol{\sigma}_M \left( AvgPool \left( \boldsymbol{P} \right) \right) \right) \right). \tag{1}$$

For average pooling, we use a kernel size of 7 and a stride of 1. In $\boldsymbol{\sigma}_M(\cdot)$, we choose $\beta = 4$ and $\gamma = 0.5$. $\boldsymbol{H}(\cdot)$ is computed according to the optical kernels from ICCAD-13 benchmark. $\boldsymbol{\sigma}_Z$ uses a threshold $I_{th} = 0.225$ and a steepness factor $\alpha = 50$. The forward pass is implemented using PyTorch builtin functions so that an SGD optimizer with a learning rate of 0.5 can be used to optimize the loss function:

$$L_f(\boldsymbol{Z}_{nom}, \boldsymbol{Z}_{max}, \boldsymbol{Z}_{max}, \boldsymbol{T}) = \|\boldsymbol{Z}_{max} - \boldsymbol{T}\|_2^2 + \|\boldsymbol{Z}_{max} - \boldsymbol{Z}_{min}\|_2^2 + L_{curv}(\boldsymbol{Z}_{nom}), \tag{2}$$

For the optimization of L2 loss, we adopt $\|\boldsymbol{Z}_{max} - \boldsymbol{T}\|_2^2$ rather than $\|\boldsymbol{Z}_{nom} - \boldsymbol{T}\|_2^2$. This technique is suggested by [3]. $\|\boldsymbol{Z}_{max} - \boldsymbol{Z}_{min}\|_2^2$ can improve the PVB. $L_{curv}(\boldsymbol{Z}_{nom}) = \sum_{x,y} (\boldsymbol{h}_{curv} \otimes \boldsymbol{Z}_{nom}(x,y))$ approximates the curvature of the mask using the mean curvature

37th Conference on Neural Information Processing Systems (NeurIPS 2023) Track on Datasets and Benchmarks.

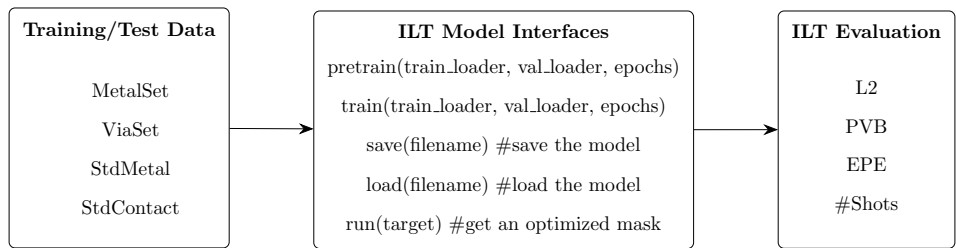

Figure 1: ILT training and evaluation flow of LithoBench.

Table 1: Comparison Between ILT Methods

| Benchmarks | ILT [5] | | | | DevelSet [6] | | | | Multi-Level [3] | | | | Ours | | | |
|---|---|---|---|---|---|---|---|---|---|---|---|---|---|---|---|---|
| | EPE | $L_2$ $(nm^2)$ | PVB $(nm^2)$ | Time (s) | EPE | $L_2$ $(nm^2)$ | PVB $(nm^2)$ | Time (s) | EPE | $L_2$ $(nm^2)$ | PVB $(nm^2)$ | Time (s) | EPE | $L_2$ $(nm^2)$ | PVB $(nm^2)$ | Time (s) |
| case1 | 6 | 49893 | 65534 | 318 | 10 | 49142 | 59607 | 1.50 | 3 | 39303 | 46077 | 1.4 | 3 | 39112 | 48831 | 6.6 |
| case2 | 10 | 50369 | 48230 | 256 | 1 | 34489 | 52010 | 1.40 | 0 | 28986 | 37626 | 1.2 | 0 | 31082 | 39102 | 6.6 |
| case3 | 59 | 81007 | 108608 | 321 | 64 | 93498 | 76558 | 1.29 | 22 | 66151 | 68021 | 1.4 | 17 | 63569 | 76183 | 6.6 |
| case4 | 1 | 20044 | 28285 | 322 | 2 | 18682 | 29047 | 1.65 | 0 | 15890 | 23511 | 1.4 | 0 | 8844 | 23986 | 6.6 |
| case5 | 6 | 44656 | 58835 | 315 | 1 | 44256 | 58085 | 0.91 | 0 | 29138 | 49987 | 1.4 | 0 | 28721 | 53856 | 6.6 |
| case6 | 1 | 57375 | 48739 | 314 | 2 | 41730 | 53410 | 0.84 | 0 | 30558 | 44503 | 1.4 | 0 | 29981 | 49084 | 6.6 |
| case7 | 0 | 37221 | 43490 | 239 | 0 | 25797 | 46606 | 0.76 | 0 | 15765 | 37009 | 1.4 | 0 | 14813 | 42364 | 6.6 |
| case8 | 2 | 19782 | 22846 | 258 | 0 | 15460 | 24836 | 1.14 | 0 | 13943 | 21503 | 0.8 | 0 | 10937 | 21210 | 6.6 |
| case9 | 6 | 55399 | 66331 | 322 | 0 | 50834 | 64950 | 1.21 | 0 | 36397 | 55600 | 1.4 | 0 | 34791 | 62161 | 6.6 |
| case10 | 0 | 24381 | 18097 | 231 | 0 | 10140 | 21619 | 0.42 | 0 | 7492 | 16604 | 1.4 | 0 | 7558 | 17393 | 6.6 |
| Average | 9.1 | 44012 | 50899 | 289 | 8 | 38402 | 48672 | **1.1** | 2.5 | 28362 | **40044** | 1.2 | **2.0** | **26941** | 43417 | 6.6 |

estimation method in [4]. The convolution kernel $\boldsymbol{h}_{curv}$ is:

$$\boldsymbol{h}_{curv} = \begin{bmatrix} -\frac{1}{16} & \frac{5}{16} & -\frac{1}{16} \\ \frac{5}{16} & -1 & \frac{5}{16} \\ -\frac{1}{16} & \frac{5}{16} & -\frac{1}{16} \end{bmatrix} \tag{3}$$

We sequentially optimize the mask at resolutions $256 \times 256$, $512 \times 512$, and $1024 \times 1024$ for 200, 100, and 100 iterations, respectively. Finally, we get the $2048 \times 2048$ mask by interpolating the result. Table 1 compares the performance of our reference ILT algorithm with SOTA ILT algorithms. It achieves the best EPE and L2 among them.

## C   Data Format

We provide the PNG images of the all data. Before being fed to DNN models, each image is divided by 255 and averaged along the channel dimension. In addition, GLP files of the target patterns are also provided. As shown in Listing 1, GLP contains two types of shapes, polygon (PGON) and rectangle (RECT). For PGON, the integer entries form a list of $(x, y)$ coordinates that represent the vertices of the polygon. The connections between adjacent vertices are horizontal or vertical. For RECT, the four integer entries are the bottom-left $(x, y)$ coordinates along with the width and height of this rectangle.

```
CELL  0OBAN_SAIL PRIME
  PGON N M1 128 128 209 128 209 263 515 263 515 344 209 344 209 479 128 479
  PGON N M1 307 614 419 614 419 496 524 496 524 812 419 812 419 695 307 695
  RECT N M1 689 321 105 315
ENDMSG
```

Listing 1: GLP Example

## D   Details of the Evaluated Models

In this section, we describe the details of the DNN models used in this paper. For all models, we use Adam [7] optimizor with a learning rate of $1 \times 10^{-3}$.

### D.1 Lithography Simulation Models

#### D.1.1 LithoGAN

LithoGAN uses a CGAN with a resolution of $256 \times 256$ to fit a lithography simulation model. The generator of the CGAN is a fully convolutional network [8] and the discriminator is a convolutional neural network (CNN). The generator consists of 8 convolutional layers and 8 transposed convolutional layers, whose kernel size is $5 \times 5$. The channel widths of the convolutional layers are listed as follows:

$$1 \rightarrow 64 \rightarrow 128 \rightarrow 256 \rightarrow 512 \rightarrow 512 \rightarrow 512 \rightarrow 512 \rightarrow 512, \tag{4}$$

where each arrow represents a layer. Each convolutional layer is followed by a $2 \times 2$ max pooling layer with a stride of 2. For the transposed convolutional layers, the channel widths are as follows:

$$512 \rightarrow 512 \rightarrow 512 \rightarrow 512 \rightarrow 512 \rightarrow 256 \rightarrow 128 \rightarrow 64 \rightarrow 2. \tag{5}$$

The discriminator of LithoGAN is a CNN consisting of 4 convolutional layers and 1 fully connected layer. The channel widths are as follows:

$$2 \rightarrow 64 \rightarrow 128 \rightarrow 256 \rightarrow 512. \tag{6}$$

Following GAN, the loss function for the discriminator is binary cross entropy, guiding the model to distinguish generated images from true images. For the generator, in addition to the loss used in GAN, LithoGAN also uses the MSE loss between the generated images and ground truths. To train LithoGAN, we use a batch size of 32. The numbers of epochs are 32 for MetalSet and 8 for ViaSet. The principle of choosing these hyperparameters is to fully utilize the memory of one NVIDIA RTX3090 GPU and train the model until it converges.

#### D.1.2 DAMO

DAMO is also based on CGAN. The generator consists of 5 convolutional layers, 9 residual convolutional layers, and 5 transposed convolutional layers, which are organized sequentially. The numbers of channels of the convolutional layers are listed as follows:

$$1 \rightarrow 64 \rightarrow 128 \rightarrow 256 \rightarrow 512 \rightarrow 1024 \tag{7}$$

The residual convolutional layers use $1024$ channels. The numbers of channels of the transposed convolutional layers are listed as follows:

$$1024 \rightarrow 512 \rightarrow 256 \rightarrow 128 \rightarrow 64 \rightarrow 2. \tag{8}$$

For all layers, the kernel size is $3 \times 3$.

The discriminator of DAMO consists of two sub-nets. One of them works on a resolution of $1024 \times 1024$. The other one downscales the input image to $512 \times 512$ before feeding it to the convolutional layers. Two sub-nets have an identical structure, containing 3 convolutional layers and 1 fully connected layer. The numbers of channels are:

$$2 \rightarrow 64 \rightarrow 128 \rightarrow 1. \tag{9}$$

The first layer is followed by a max pooling layer. For all layers, the kernel size is $4 \times 4$. We use the same loss function as LithoGAN for DAMO. To train DAMO, we use a batch size of 4. The numbers of epochs are 8 for MetalSet and 2 for ViaSet.

#### D.1.3 DOINN

Inspired by Fourier Neural Operator (FNO) [9], DOINN utilizes a novel reduced FNO (RFNO) architecture to fit the lithography simulation model. An RFNO layer is defined as:

$$\boldsymbol{F}_R(\boldsymbol{X}) = \sigma\left(\mathcal{F}^{-1}\left(\mathcal{F}(\boldsymbol{X}) \otimes \boldsymbol{W}_P \cdot \boldsymbol{W}_R\right)\right), \tag{10}$$

where $\mathcal{F}$ and $\mathcal{F}^{-1}$ represent FFT and inverse FFT, respectively. $\boldsymbol{W}_P$ is a complex-valued weight matrix with a size of $1 \times C \times 1 \times 1$. In this paper, we use $C = 64$. $\boldsymbol{W}_R$ is another complex-valued weight matrix with the same size as the layer input. The sigmoid function is represented by $\sigma$.

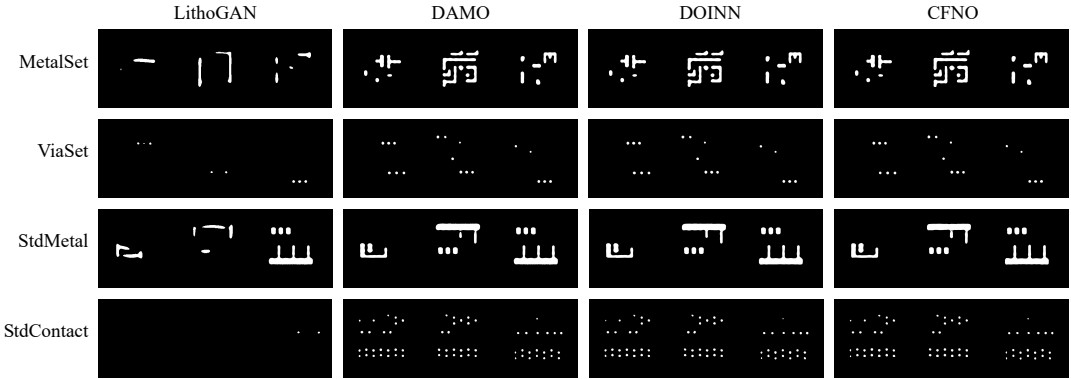

Figure 2: Samples of lithography simulation.

DOINN includes two branches, the global perception branch, and the local perception branch. The former branch consists of an $8 \times 8$ average pooling layer and an RFNO layer. The latter branch consists of three convolutional layers. The numbers of channels are:

$$1 \rightarrow 16 \rightarrow 32 \rightarrow 64. \tag{11}$$

The outputs of two branches are concatenated and then fed to 6 convolutional layers. The numbers of channels are:

$$128 \rightarrow 64 \rightarrow 32 \rightarrow 16 \rightarrow 16 \rightarrow 8 \rightarrow 1. \tag{12}$$

The first three layers are followed by $2 \times 2$ upscaling layers. To train DOINN, we use a batch size of 16. The numbers of epochs are 32 for MetalSet and 8 for ViaSet.

### D.1.4  CFNO

A Convolutional Fourier Neural Operator (CFNO) layer includes the following steps. The input image is split into $k \times k$ patches. After that, each patch are processed by the following operations:

$$\boldsymbol{F}_C(\boldsymbol{X}) = \sigma\left(\mathcal{F}^{-1}\left(\mathcal{F}(\boldsymbol{X}) \cdot \boldsymbol{W}_C\right)\right). \tag{13}$$

Finally, we apply the token-wise convolution operation which is implemented by a separable convolution layer with a kernel size of $3 \times 3$.

To encode the input image, the complete CFNO network uses 4 branches. Three of them are CFNO layers, with $k = 16$, $k = 32$, and $k = 64$. The 4th branch includes 9 successive $3 \times 3$ convolutional layers, whose channel widths are 32, 64, and 128 (3 layers for each width). The outputs of the branches are concatenated to form the encoded features.

The decoding flow includes 12 convolutional layers, split into 4 groups. The layers in each group share the same number of channels. The channel widths of the groups are:

$$128 \rightarrow 64 \rightarrow 32 \rightarrow 32. \tag{14}$$

Each group in the first three contains a $2 \times 2$ upscaling layer. Finally, a convolutional layer with 2 channels outputs the predicted results. To train CFNO, we use a batch size of 4. The numbers of epochs are 8 for MetalSet and 2 for ViaSet.

Fig. 2 presents some examples from the tested models. Except for LithoGAN, the outputs from other models are visually similar, which is consistent with the quantitive results.

### D.2  Mask Optimization Models

### D.2.1  GAN-OPC

GAN-OPC also follows the design of CGAN. The generator consists of 5 convolutional layers and 5 pixel-shuffle [10] convolutional layers layer. The channel widths are listed as follows:

$$1 \rightarrow 16 \rightarrow 64 \rightarrow 128 \rightarrow 512 \rightarrow 1024 \rightarrow 512 \rightarrow 128 \rightarrow 64 \rightarrow 16 \rightarrow 1. \tag{15}$$

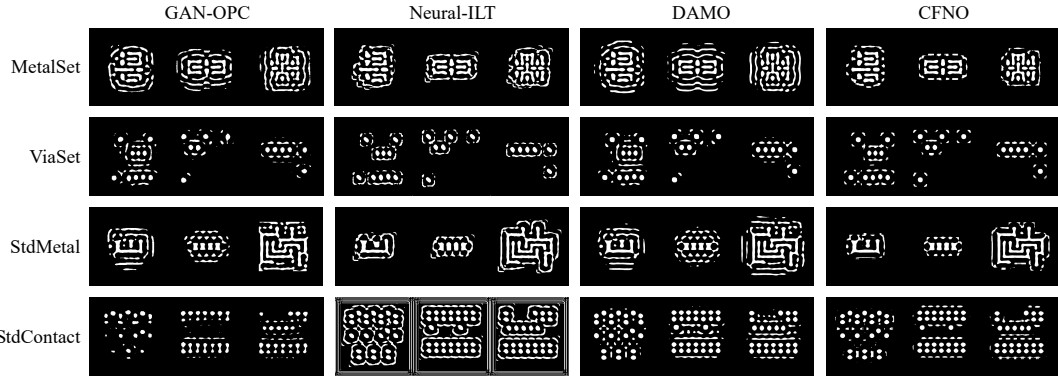

Figure 3: Samples of mask optimization.

The discriminator consists of 15 convolutional layers and 3 fully connected layers. The convolutional layers are divided into 5 groups. The channel widths of the groups are:

$$1 \rightarrow 64 \rightarrow 128 \rightarrow 256 \rightarrow 512 \rightarrow 512. \tag{16}$$

The fully connected layers have the following sizes:

$$32768 \rightarrow 2048 \rightarrow 512 \rightarrow 1. \tag{17}$$

GAN-OPC involves two training stages. In the first stage, the generator is trained to minimize the MSE between the generated images and the reference optimized masks. In the second stage, it incorporates a lithography-guided loss function along with the GAN training process. Specifically, we use the L2 loss as an additional objective, where the printed image $\boldsymbol{Z}_{nom}$ is obtained by the reference lithography simulation model. To train GAN-OPC, we use a batch size of 64. The numbers of epochs are 64 for MetalSet and 16 for ViaSet.

### D.2.2 Neural-ILT

Neural-ILT consists of 8 convolutional layers and 8 transposed convolutional layers. Every 2 layers are grouped together. The channel widths are:

$$1 \rightarrow 64 \rightarrow 128 \rightarrow 256 \rightarrow 512 \rightarrow 256 \rightarrow 128 \rightarrow 64 \rightarrow 1. \tag{18}$$

Following UNet [11], skip connections are added to the model. Specifically, the features of the layers that have the same size are concatenated before being fed to the next layer.

The training of Neural-ILT also consists of two stages. The first stage minimizes the MSE between the generated images and the reference optimized masks. The second stage optimizes $L2(\boldsymbol{Z}_{nom}, \boldsymbol{T}) + PVB(\boldsymbol{Z}_{max}, \boldsymbol{Z}_{min})$. To train Neural-ILT, we use a batch size of 12. The numbers of epochs are 16 for MetalSet and 4 for ViaSet.

### D.2.3 DAMO

DAMO for mask optimization adopts $L1(\boldsymbol{Z}_{nom}, \boldsymbol{T}) + PVB(\boldsymbol{Z}_{max}, \boldsymbol{Z}_{min})$ at the second training stage. *L1* is the Manhattan distance. Other details are similar to the DAMO for lithography simulation. To train DAMO, we use a batch size of 4. The numbers of epochs are 8 for MetalSet and 4 for ViaSet.

### D.2.4 CFNO

CFNO for mask optimization shares the same structure as the CFNO for lithography simulation. The training process minimizes the distance between the generated masks and the reference masks. At each training step, we compare the L2 loss of a generated mask and its corresponding reference mask. If the L2 of the generated mask is better, the training on this datum is skipped. To train CFNO, we use a batch size of 4. The numbers of epochs are 8 for MetalSet and 2 for ViaSet.

Fig. 3 presents some examples from the tested models. Compared to GAN-OPC, DAMO outputs more regular shapes, while CFNO gets less complex patterns. Although Neural-ILT achieves the best performance on StdContact, its outputs contain some weird lines, which should be avoided in future mask optimization models.

Table 2: Comparison on Finetuned ILT Results

| Subtask | GAN-OPC [12] $L_2$ | PVB | EPE | Shots | Neural-ILT [13] $L_2$ | PVB | EPE | Shots | DAMO [14] $L_2$ | PVB | EPE | Shots | CFNO [15] $L_2$ | PVB | EPE | Shots |
|---------|------|------|------|------|------|------|------|------|------|------|------|------|------|------|------|------|
| 1 | **27091** | 43168 | 2.0 | 552 | 27407 | **42764** | 2.6 | 547 | 27300 | 43227 | **1.8** | 551 | 27608 | 42888 | 2.7 | **524** |
| 2 | 5359 | 9447 | **0.2** | 287 | **5131** | **9343** | 0.2 | 309 | 5603 | 9486 | **0.2** | **279** | 5515 | 9449 | **0.2** | 283 |
| 3 | 12841 | 24859 | **0.0** | 441 | **12700** | **24773** | **0.0** | 450 | 12883 | 24956 | **0.0** | 442 | 12957 | 24999 | 0.1 | **422** |
| 4 | 31223 | **41339** | 8.2 | **627** | 27559 | 42819 | 4.4 | 700 | 27910 | 43651 | **3.6** | 640 | 28053 | 43363 | 4.0 | 641 |
| Average | 19128 | **29703** | 2.6 | 476 | **18199** | 29924 | 1.8 | 501 | 18424 | 30330 | **1.4** | 478 | 18533 | 30174 | 1.7 | **467** |

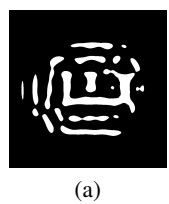 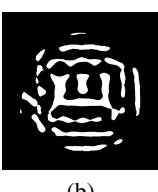 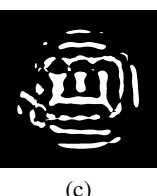 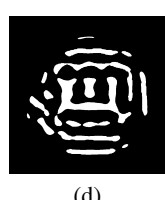 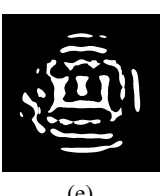

(a)          (b)          (c)          (d)          (e)

Figure 4: Samples of finetuned mask optimization. (a) Reference; (b) GAN-OPC; (c) Neural-ILT; (d) DAMO; (e) CFNO.

## E Finetuned Results

In typical DNN-based ILT methods, the output masks from DNN models can be finetuned by traditional ILT methods to get better results. In this paper, we use the reference ILT algorithm to finetune the masks from the tested models. Table 2 compares the finetuned results. Finetuning bridges the huge gaps between different methods. Nevertheless, each method still has its own strengths and benefits. GAN-OPC achieves the best PVB. Neural-ILT keeps the lowest L2 loss. CFNO obtains the smallest shot counts. Although the superiority of DAMO is not so significant, it achieves an impressive EPE score. Fig. 4 shows some examples of the finetuned results. The difference between the masks from different models is not as large as it was before finetuning.

## F Limitations and Future Work

Although DAMO [14], DOINN [16], and AdaOPC [17] have pushed forward large-scale ILT for simple patterns, the optimization for more production-level scale and complex layout patterns has not been well studied. Thus, LithoBench has not included a super large-scale evaluation for DNN-based lithography simulation and mask optimization models. However, this will not affect the quality of the dataset and hence the efficacy of benchmarking AI lithography solutions, because of two reasons. Firstly, the patterns that appear on one layer are somewhat similar because they are created through standard chip physical design flows from EDA vendors. Secondly, ILT is typically performed in a tile-based manner in real semiconductor foundries, due to limitations of computing resources [18]. Therefore, containing such a number of tile-based data in our benchmark suite is proper for a comprehensive evaluation of the models. In future work, LithoBench can evolve to support more production-level ILT based on the future progresses of corresponding research.

We hope LithoBench and our experimental results can contribute to the further development of computational lithography. Since we use circuit layouts that have no personally identifiable information or offensive content, users can be free to use LithoBench in their research.