# OpenReview forum: "LithoBench: Benchmarking AI Computational Lithography for Semiconductor Manufacturing"
_NeurIPS.cc/2023/Track/Datasets_and_Benchmarks — NeurIPS 2023 Datasets and Benchmarks Poster_

### Official Review · Reviewer_zijQ · 2023-07-19
**OK work but questionable/limited applicability.**

**Rating:** 6
**Confidence:** 3
**Correctness:** No issue with correctness.
**Clarity:** The writing quality of the paper is g…

**Strengths:**

- Addresses the need for rigorous benchmarks for deep learning based lithography methods with diverse, realistic data
- Includes both lithography simulation and mask optimization tasks
- Tests several recent works on the benchmark and provides analysis and insights
- Might be the first dataset/benchmark on DNN-based lithography simulation and mask optimization, but I'm not sure, the authors should clarify this.

**Additional Feedback:**

More discussions about the actual applicability of this benchmark for future DNN-based lithography simulation and mask optimization works given the lack of full-chip data is appreciated and will help me to better understand the contribution and limitation of this work.

**Documentation:**

No issue with documentation.

**Ethics:**

No issue with ethics.

**Limitations:**

I appreciate the authors for acknowledging the limitation of lacking full-chip scale data in the limitation section in supplementary material and mentioning it could be considered in future work. But to be honest, this is my biggest concern after reading the paper. The authors mentioned that the main limitation of traditional ILT methods is the computational overhead, making it challenging to implement ILT at full-chip scale, hence DNN-based ILT methods are proposed to overcome this limitation. Hence, at least one of the target use cases for DNN-based ILT methods is full-chip ILT, but this is not provided in the current dataset, even very simple full-chip patterns.

I'm not an expert in lithography optimization and hence not tracking the recent works in this field, but for me, the value of this dataset is questionable/limited for future DNN-based lithography simulation or mask optimization works given the lack of full-chip data, which is probably the ultimate goal of DNN-based methods.

**Opportunities For Improvement:**

- Different sets (metal, via, stdmetal, stdvia) seem to not use the same tech node, with metal seeming to use 32nm while stdmetal and via use Nangate 45nm. The authors also do not clearly list the exact tech nodes used. Mixing tech nodes might not be good since the lack of consistency in nodes makes it harder to draw node-specific conclusions or optimize models targeting a single node.

- Are there any existing datasets/benchmarks for DNN-based ILT? The paper does not mention or compare to existing lithography datasets. If this is the first dataset for DNN-based ILT, please make it clear in the paper as this will be a unique contribution, so future DNN-based ILT works have a standard for evaluation and comparison. If there are existing datasets, then comparisons should be made in the paper and highlight the differences and advantages of this work.

**Relation To Prior Work:**

Please see my comments in Opportunities For Improvement part.

**Summary And Contributions:**

The paper presents LithoBench, a new dataset and benchmark for training and evaluating deep learning based lithography simulation and mask optimization. This work could potentially set a standard for evaluating and comparing future DNN-based ILT works.

---

> ### Author Response · Authors · 2023-08-14
>
> (1) Different sets (metal, via, stdmetal, stdvia) seem to not use the same tech node, that hard to draw node-specific conclusions.
>
> The authors appreciate the comments and we apologize for the ambiguity.
> Technology nodes (especially adjacent generations like 32nm and 45nm) do not necessarily reflect lithography behavior or properties, which are most likely affected by design critical dimension (metal width, Via width and height, etc) and pitch (spacing between shapes).
> Here, all benchmark groups MetalSet, ViaSet, StdMetal, and StdContact have similar design critical dimensions (65nm-100nm) and pitch (50nm-80nm) sizes.
> And all our benchmark groups are developed based on a simulator targeting 193nm DUV lithography, which is applicable to designs with such critical dimensions and pitch sizes.
> Therefore, technology node difference in our benchmark sets does not affect the quality of the benchmark and will not prevent users draw any node-specific conclusions through our dataset.
>
> It should also be noted that MetalSet and ViaSet are related to inter-standard cell circuit connection layouts, while the StdMetal and StdContact are related to intra-standard cell layouts.
> Inter- and intra-standard cell layouts usually have different design styles (topology, shape density, and so on), including both will make a more complete benchmark suite for lithography applications.
> This is a motivation including both all four benchmark groups.
>
>
> (2) Are there any existing datasets/benchmarks for DNN-based ILT?
>
> Technically, there are no DNN-based ILT benchmarks with good quality in the literature.
> To the best of our knowledge, the only related dataset is related to GAN-OPC [*1], where around 4k low-resolution images are synthesized by shuffling shapes in the original 10 ICCAD13 clips.
> And the ground truth results are generated by a legacy ILT engine that has a significant performance gap compared to SOTA.
> We have added Section 2.4 to compare LithoBench with related works.
> We analyzed their weaknesses in Section 2.4 and show that LithoBench is the first comprehensive dataset that simultaneously supports lithography simulation and mask optimization.
>
>
> (3) The value of this dataset is questionable/limited for future DNN-based lithography simulation or mask optimization works given the lack of full-chip data.
>
> We apologize for causing concerns and misunderstandings. By full-chip design, we mean a more production-level case for benchmarking purposes which is expected to be more complex than our current version. We have revised the term in Section 1 and Appendix F to avoid ambiguity.
> Also, it is not necessary to have a super-large/full chip design to benchmark the performance of lithography solutions, because lithography simulation and mask optimization are typically performed in a tile-based manner in real semiconductor foundries, due to limitations of computing resources [*2].
> Therefore, containing only tile-based data in our benchmark suite will not affect the quality of the dataset and hence the efficacy of benchmarking AI lithography solutions.
> We also add a related discussion in Appendix F.
>
>
> Reference
>
> [*1] H. Yang et al., “GAN-OPC: Mask optimization with lithography-guided generative adversarial nets,” in ACM/IEEE Design Automation Conference (DAC), 2018.
>
> [*2] V. Singh et al., “Making a trillion pixels dance,” Optical Microlithography XXI. Vol. 6924. SPIE, 2008.

---

> > ### Comment · Reviewer_zijQ · 2023-08-18
> >
> > Thanks for the response and revision regarding my comments. After reading it, I decide to raise my rating to 6.

---

### Official Review · Reviewer_uJf3 · 2023-07-21
**A Benchmark for Lithography Task in Semiconductor Manufacturing**

**Rating:** 6
**Confidence:** 3
**Correctness:** All claims seem to be correct.

**Strengths:**

The data about lithography can be very sensitive due to its importance in the semiconductor manufacturing industry, making it hardly available to many users. Releasing this will be helpful to AI research in this direction.

**Additional Feedback:**

N/A

**Clarity:**

Some personal suggestions in writing.
- Better make the definition of lithography simulation and mask optimization more obvious/clear in Section I. Are they both sub-steps of the traditional ILT process? This is not obvious to readers without background.
- What specifically is ILT algorithm in Section 2.2? Is it about lithography simulation or mask optimization or others?
- In Figure 1, better increase the distance between (a) and (b). They are too close right now.
- The main difference/relationship between the MetalSet and ICCAD-13 [21] is not very clear at first glance.
- How is the Ref. in Figure 4 & 5 generated? Can it be viewed as ground-truth?
- Better provide a reference to the GLP format. Any open-source parsers for this data format?

**Documentation:**

The documentation seems to be sufficient.

**Ethics:**

No ethics concerns.

**Limitations:**

The writing can be improved to be more clear. Many concepts/terms may be obvious to the authors, but lithography is a complex task for readers without a rich background. Detailed suggestions are given in the 'Clarity' part.

**Opportunities For Improvement:**

The authors better explicitly summarize/mention all prior benchmarks/datasets in this ILT or lithography task. Currently, all prior works are mixed in Section 2, most of them are methodologies instead of prior datasets.

The authors better slightly elaborate on why this dataset is representative and comprehensive enough for fair benchmarking.

The title mentions 'LithoBench and lithography', but the benchmark is only about ILT. Actually, there are other benchmarks and studies for lithography hotspot detection. Perhaps either mention it in the preliminary or revise the title?

**Relation To Prior Work:**

- As mentioned above, the authors better explicitly mention all prior benchmarks/datasets in this ILT or lithography task. Currently all prior works are mixed in Section 2, most of them are methodologies instead of prior datasets.

**Summary And Contributions:**

This work proposes a collection of circuit layout tiles for deep-learning-based lithography simulation and mask optimization. LithoBench consists of more than 120k tiles that are cropped from real circuit designs or synthesized according to the layout topologies of famous ILT testcases.

---

> ### Author Response · Authors · 2023-08-14
>
> (1) The authors better explicitly summarize/mention all prior benchmarks/datasets in this ILT or lithography task.
>
> Thanks for your comment. We have provided the comparison between LithoBench and existing benchmarks in Section 2.4.
> LithoBench is the first comprehensive dataset that simultaneously supports lithography simulation and mask optimization.
> Another advantage is that LithoBench includes abundant and diverse data that covers synthetic and real-world layout tiles.
>
>
> (2) The authors better slightly elaborate on why this dataset is representative and comprehensive enough for fair benchmarking.
>
> Thanks for your comment. As discussed in the new Section 2.4 and the 5th paragraph in Section 1, existing DNN-based ILT methods may focus on metal or via layer, synthetic or real-world data.
> However, the existing benchmark cannot cover all of the scenarios.
> ICCAD-13 provides 10 samples for metal-layer mask optimization. GAN-OPC provides 4k data, which are random combinations of existing patterns in ICCAD-13.
> They cannot provide data for lithography simulation and via-layer mask optimization. Furthermore, they cannot offer ground truth results that are comparable to SOTA.
> LithoBench simultaneously supports lithography simulation and mask optimization. It can cover metal and via layers, as well as synthetic and real-world data, with SOTA ground truth results.
> In addition, LithoBench provides much more data than existing benchmarks. Thus, it is more representative and comprehensive.
>
>
> (3) Actually, there are other benchmarks and studies for lithography hotspot detection. Perhaps either mention it in the preliminary or revise the title?
>
> Thanks for your comment. Lithography hotspot detection only indicates the presence of defects in certain regions of the printed patterns, but provides little information for mask optimization.
> Lithography simulation produces can provide detailed information about the printed patterns, which is more useful than hotspot detection.
> That is why hotspot detection is not included in LithoBench.
> We have included related discussions in Section 2.4
>
> (4) Better make the definition of lithography simulation and mask optimization more obvious/clear in Section I.
>
> Thanks for your comment. We have improved the definition of lithography simulation and mask optimization in the 7th paragraph in Section 1.
> In simple terms, the output of ILT is an optimized mask, which can get similar results as the target patterns after the lithography processes.
> DNN-based lithography simulation approximates the lithography processes with DNN models, which can provide a fast evaluation of the mask quality.
> DNN-based mask optimization models directly obtain the optimized mask in one forward pass.
>
>
> (5) What specifically is ILT algorithm in Section 2.2? Is it about lithography simulation or mask optimization or others?
>
> Thanks for your comment. It is mask optimization. We have changed the title of Section 2.2 to "Mask Optimization via ILT".
>
>
> (6) In Figure 1, better increase the distance between (a) and (b). They are too close right now.
>
> Thanks for the correction. We have increased the distance between Fig.1(a) and Fig.1(b).
>
>
> (7) The main difference/relationship between the MetalSet and ICCAD-13 [21] is not very clear at first glance.
>
> Thanks for your comment. We have improved the description of MetalSet in Section 3.1.1.
> We design MetalSet to enable the training of a DNN-based model that can achieve high performance on the ICCAD-13 benchmark.
> Each tile in MetalSet is randomly generated following the design rule of the ICCAD-13 benchmark.
> The original ICCAD-13 benchmark is used to evaluate DNN-based mask optimization models that are trained on MetalSet.
>
>
> (8) How is the Ref. in Figure 4 & 5 generated? Can it be viewed as ground truth?
>
> Yes. They are ground truths. We have changed "Ref." to "Ground truth" in Fig.4 and Fig.5.
>
>
> (9) Better provide a reference to the GLP format. Any open-source parsers for this data format?
>
> Thanks for your comment. We have provided the reference in Section 3.1.1.
> An example of GLP file is shown in Appendix C.
> GLP is a simple text format that can be parsed with a few lines of code.
> The major content of a GLP file is the nodes of polygon shapes.

---

> > ### Comment · Reviewer_uJf3 · 2023-08-20
> >
> > Thanks for your comments, and the writing is better from my perspective. I raised the score to 6.
> >
> > Another minor question, you mentioned in Section 3.1.1: "We synthesize 16,472 tiles using the layout generation method in [48]. Each tile is randomly generated following the design rule of ICCAD-13 benchmark. The original ICCAD-13 benchmark is used to evaluate DNN-based mask optimization models that are trained on MetalSet." Will it lead to information leakage between training and testing dataset?

---

> > > ### Author Response · Authors · 2023-08-22
> > >
> > > (1) Each tile in MetalSet is randomly generated following the design rule of ICCAD-13 benchmark. Will it lead to information leakage between training and testing dataset?
> > >
> > > The authors are very grateful for your valuable comments.
> > > Technically, the answer to this question is no. In semiconductor manufacturing, the "design rule" is a general term defined by some specific technology node (e.g. Intel 14nm, TSMC N7, etc.).
> > > To be manufactured under a certain technology node, all designs must follow the same design rules, including the minimum width of a shape, the minimum distance between different shapes, the minimum/maximum area of a shape, etc.
> > > These rules are the premises of satisfactory the yield of the technology node.
> > > Therefore, the training and the testing sets are supposed to share the same design rules to make the benchmarking and ML applications meaningful.
> > > It should also be noted that though the training and testing instances share the same design rules, they are still different designs with different shape topologies, shape areas, connectivities and etc. These differences will be the major objectives that AI models should be generalized on.
> > > In conclusion, design rule-based random generation won't pose concerns about information leakage between the training and testing set.
> > >
> > > To make the manuscript more clear, we have included related discussions in Sec 3.1.1.

---

### Official Review · Reviewer_EsWu · 2023-07-23
**Paper seems solid, I need help understanding context and more marketing**

**Rating:** 8
**Confidence:** 4
**Correctness:** I think so?
**Clarity:** I think so.

**Strengths:**

This appears to be the first large-scale lithography dataset for mask optimization and simulation using real world designs/layouts.

**Additional Feedback:**

I really want to understand the significance of your dataset. Is this the first such dataset? Are there existing datasets for this purpose?

I think this paper is probably quite significant, but you need to help me understand it better. More context around the dataset and the novelty and comparison to existing efforts would help a ton.

If you can help me on the above points I will improve my score.

Update: I changed my score.

**Documentation:**

Their dataset site looks good.

**Ethics:**

Seems fine.

**Limitations:**

Yes. There are no negative implications of this work.

**Opportunities For Improvement:**

Help me understand the dataset and its novelty more!

Update: I think the paper is more clear, raising rating.

**Relation To Prior Work:**

No. Please compare to prior works. Or let us know that no such thing exists and this is super novel!

Update: Authors have significantly improved the related work section. Thanks!

**Summary And Contributions:**

Lithography is super cool and this is doing work to make it better. ILT is very important in semiconductor design and manufacturing. So this is a very important problem area. The authors have created a great dataset (i think?) and done some initial work on it.

A lot of the content focuses on lithography rather than the dataset itself, which makes it tough for me.

Update: This appears to be the first large-scale lithography dataset for mask optimization and simulation using real world designs/layouts. I think this is an especially valuable area and am very excited by the dataset hence the rating.

---

> ### Author Response · Authors · 2023-08-14
>
> (1) Help me understand the dataset and its novelty more!
>
> Thanks for your comment. We have provided the comparison between LithoBench and existing benchmarks in Section 2.4.
> Technically, there are no DNN-based ILT benchmarks with good quality in the literature.
> To the best of our knowledge, the only related datasets are ICCAD-13 and GAN-OPC [*1], designed for mask optimization.
> ICCAD-13 has only 10 samples, which are not suitable for deep learning.
> GAN-OPC has around 4K low-resolution images, but the size of dataset is still unsatisfactory, and the ground truth results are generated by a legacy engine that has a significant performance gap compared to SOTA.
> LithoBench is the first comprehensive dataset that simultaneously supports lithography simulation and mask optimization.
> Another advantage is that LithoBench includes abundant and diverse data that covers synthetic and real-world chip layout tiles.
>
>
> Reference
>
> [*1] H. Yang et al., “GAN-OPC: Mask optimization with lithography-guided generative adversarial nets,” in ACM/IEEE Design Automation Conference (DAC), 2018.

---

> > ### Comment · Reviewer_EsWu · 2023-08-19
> > **Thanks!**
> >
> > This is fairly helpful. I would suggest adding a stronger sentence somewhere in the paper. E.g., you write "LithoBench consists of more than 120k tiles that are cropped from real circuit designs or synthesized according to topologies of widely adopted ILT testcases."
> >
> > For example, could you say that this is the first large scale lithography dataset built from real world designs? Or maybe it's the first lithography dataset using real world designs of sufficient size for training?
> >
> > Anyway, appreciate the related work and will increase my rating!

---

> > > ### Author Response · Authors · 2023-08-22
> > >
> > > (1) I would suggest adding a stronger sentence somewhere in the paper.
> > >
> > > We greatly appreciate your positive feedback and valuable suggestions.
> > > We have emphasized at the end of Section I that LithoBench is the first large-scale dataset for computational lithography built from real-world designs.

---

### Official Review · Reviewer_hRPs · 2023-07-23
**A useful dataset and solid benchmarking for semiconductor manufacturing, needs improvement on related works**

**Rating:** 7
**Confidence:** 2
**Correctness:** I am not an expert in semiconductor m…
**Clarity:** Yes

**Strengths:**

1. The paper is well-written, the section 2 does a good job providing reader the background about lithography simulation and ILT, making the paper much friendly to people who are not familiar with semiconductor manufacturing process.
2. The observations drawn from benchmarking the recent methods can be useful for the community.

**Additional Feedback:**

None

**Documentation:**

Yes

**Limitations:**

I would suggest the author to also discuss the limitation of the dataset.

**Opportunities For Improvement:**

1. I would suggest the paper also provides a dedicated related work section to compare the proposed dataset with other dataset for this problem. For example, I am not very clear what's the comparison between the proposed one and ICCAD-13 benchmark, even the paper has mentioned the later one in several places.
2. Due to missing the related work section, I also can't figure out whether all the proposed evaluation metrics in section 3.3.2 are all original, or already introduced by other prior works.
3. It would be better to also summarize the statistics of the proposed dataset in a table of figure in Section 3. Right now, all the details of the dataset are buried in the text.

**Relation To Prior Work:**

Needs improvement as discussed in above section 'opportunities for improvement'.

**Summary And Contributions:**

1. The paper presents a new dataset, which contains >100K tiles, to facilitate the development and benchmarking of the ML-based lithography simulation and mask optimization.
2. The paper proposes a set of metrics to evaluate the ML-based models's performance for this problem.
3. The paper also evaluated some recent lithography simulation and mask optimization methods, and draw some useful findings.

---

> ### Author Response · Authors · 2023-08-14
>
> (1) Provides a dedicated related work section. Show the comparison between the proposed one and ICCAD-13 benchmark.
>
> Thanks for your advice. We have provided the comparison between LithoBench and existing benchmarks in Section 2.4.
> ICCAD-13 is a famous benchmark for mask optimization.
> However, it is targeting numerical optimization solutions only and the 10 instances are too small to fit AI solutions.
>
>
> (2) The metrics in section 3.3.2 are all original, or already introduced by other prior works?
>
> Thanks for your comment. These metrics come from existing works, and are well accepted in industry [*1]. We have added the reference in Sections 3.3.1 and 3.3.2.
>
>
> (3) Summarize the statistics of the proposed dataset in a table.
>
> Thanks for your suggestion. We have summarized the statistics in Table 1.
>
>
> (4) Discuss the limitation of this work.
>
> Thanks for your advice. We have put the discussion in Appendix F.
> A limitation is that LithoBench has not included a super large-scale evaluation for DNN-based lithography simulation and mask optimization models.
> However, this will not affect the quality of the dataset and hence the efficacy of benchmarking AI lithography solutions, because of two reasons.
> Firstly, the patterns that appear on one layer are somewhat similar because they are created through standard chip physical design flows from EDA vendors.
> Secondly, lithography simulation and mask optimization are typically performed in a tile-based manner in real semiconductor foundries, due to limitations of computing resources [*2].
> Therefore, a proper number of tile-based data are enough for a comprehensive evaluation of the models.
>
>
> Reference:
>
> [*1] Pang, Linyong, et al. "Study of mask and wafer co-design that utilizes a new extreme SIMD approach to computing in memory manufacturing: full-chip curvilinear ILT in a day." Photomask Technology 2019. Vol. 11148. SPIE, 2019.
>
> [*2] V. Singh et al., “Making a trillion pixels dance,” Optical Microlithography XXI. Vol. 6924. SPIE, 2008.

---

### Official Review · Reviewer_URA3 · 2023-07-26
**The proposed dataset is a promising dataset for benchmarking platform of AI-driven computational lithography, which is valuable and deserve an accept.**

**Rating:** 7
**Confidence:** 3
**Clarity:** The paper appears to be reasonably we…

**Strengths:**

- The research is closely tied to real-world applications in the semiconductor industry, addressing a significant challenge in this field - the computational burden of ILT.
- The LithoBench dataset represents a comprehensive collection of real and synthetic circuit design tiles, which fills a significant gap in the field by providing a large-scale, diverse, and practical dataset for training and evaluating deep learning models for ILT.
- The LithoBench dataset is openly available, which can stimulate further research and model development in the community.

**Additional Feedback:**

None

**Correctness:**

The authors have taken steps to make the dataset and benchmarking valuable to the community. They have constructed the LithoBench dataset from real-world and synthetic circuit designs, which should provide a broad range of test cases. They have also implemented a benchmarking platform that evaluates various metrics like quality, robustness, complexity, and runtime, which is a comprehensive approach to assessing the performance of different methods. So the claims made in the submission are correct.

**Documentation:**

There are several aspects of the dataset's creation and maintenance that are not explicitly addressed in the provided text. For example, there's no specific information about the dataset's licensing, hosting, or long-term maintenance plan. Nor is there any mention of ethical considerations or intended uses of the dataset.

**Ethics:**

None.

**Limitations:**

The effectiveness of the proposed deep learning models for lithography simulation and mask optimization is heavily dependent on the quality and diversity of the LithoBench dataset. If the dataset lacks diversity or does not adequately cover all possible circuit design scenarios, the developed models may not be well-suited for real-world applications.

**Opportunities For Improvement:**

The effectiveness of the developed models is heavily reliant on the quality and diversity of the LithoBench dataset. If the dataset is not comprehensive enough, the models may not be well-trained for all possible scenarios.
The paper discusses training and evaluating deep learning models using the LithoBench dataset which is based on simulation tools, it'd be more helpful to provide clear evidence of how these models perform in real-world, industrial-scale applications.


**Relation To Prior Work:**

The authors acknowledge that various deep neural network (DNN)-based Inverse Lithography Technology (ILT) methods have been proposed, which could boost speed and performance of ILT. However, they note the difficulty in comparing these methods comprehensively due to the differing scenarios and training data they were evaluated on. They also mention the challenge of applying them in production.
The authors then position their contribution, LithoBench, as a comprehensive dataset and benchmarking platform that can evaluate different DNN ILT methods in various scenarios. They argue that this comprehensive and versatile benchmarking platform has not been previously available, making LithoBench a unique and significant contribution to the field.

**Summary And Contributions:**

This paper focuses on the development and evaluation of AI and machine learning methods for inverse lithography technology (ILT), which is a critical step in semiconductor manufacturing. The authors introduce LithoBench, a large-scale dataset (consisting of more than 120,000 tiles cropped from real circuit designs and synthesized according to the layout topologies of known ILT test cases) of real and synthetic circuit designs for training and evaluating deep learning models in lithography simulation and mask optimization. These models, which reduce the computational complexity and time required by ILT, can be assessed on the LithoBench platform using various metrics, including quality, robustness, complexity, and runtime. This work is intended to accelerate the development of AI-driven computational lithography.

---

> ### Author Response · Authors · 2023-08-14
>
> (1) It'd be more helpful to show how these models perform in real-world, industrial-scale applications.
>
> Thanks for your advice. Due to the high-cost manufacturing process, simulation-based methods are also adopted in industry [*1].
> Moreover, lithography simulation and mask optimization are typically performed in a tile-based manner in real semiconductor foundries, due to limitations of computing resources [*2].
> Therefore, the simulation-based and tile-based schemes in our benchmark suite will not affect the quality of the dataset and hence the efficacy of benchmarking AI lithography solutions.
> We also add a related discussion in Appendix F.
> Additionally, both StdMetal and StdVia are industrial standard designs, which should be effectively reflecting the AI model performance practically.
>
>
> (2) If the dataset lacks diversity or does not adequately cover all possible circuit design scenarios, the developed models may not be well-suited for real-world applications?
>
> Thanks for your comment.
> Typically AI for lithography solutions is developed layer by layer and design by design, to make the model training easier.
> Though different layers or different chip designs are most likely to show distribution shifts,  they usually have similar lithography behavior.
> Therefore, benchmarking AI models on a certain layer and certain design will largely reflect the behavior of the AI mode in real-world applications.
>
> Given a technology node, the patterns that appear on one layer are somewhat similar because they are created through standard chip physical design flows from EDA vendors (e.g., Cadence Innovus or Synopsys IC Compliler).
> LithoBench covers metal and via layers, which are representative and complex enough for evaluating lithography simulation and mask optimization models.
>
>
> Reference
>
> [*1] Pang, Linyong. "Inverse lithography technology: 30 years from concept to practical, full-chip reality." Journal of Micro/Nanopatterning, Materials, and Metrology 20, no. 3, 2021
>
> [*2] V. Singh et al., “Making a trillion pixels dance,” Optical Microlithography XXI. Vol. 6924. SPIE, 2008.

---

> > ### Comment · Reviewer_URA3 · 2023-08-22
> >
> > Most of my concerns have been addressed. Thank you for the detailed experiments on various platforms. I'll maintain my score.

---

### Decision · Program_Chairs · 2023-09-22

**Decision:**

Accept (Poster)

**Comment:**

This paper the needs of dataset for using machine learning techniques for for inverse lithography technology (ILT), which is a critical step in semiconductor manufacturing. Given the high computational complexity, ML based solutions can be attractive.  The authors introduce LithoBench, a large-scale dataset (consisting of more than 120,000 tiles for different layers, including metal set, via set, standard cell metal set, and standard cell contact set). It is a good contribution to the community.  There is sufficient discussions between the reviewers and the authors, which help to improve the quality of the manuscript.